# Amphioxus adenosine-to-inosine tRNA-editing enzyme that can perform C-to-U and A-to-I deamination of DNA

Zhan Gao [1,4 ✉], Wanyue Jiang[2,4], Yu Zhang[1,4], Liping Zhang[1], Mengmeng Yi[1], Haitao Wang[1], Zengyu Ma[1], Baozhen Qu[1], Xiaohan Ji[1], Hongan Long[2,3] & Shicui Zhang [1,3 ✉]

Adenosine-to-inosine tRNA-editing enzyme has been identified for more than two decades, but the study on its DNA editing activity is rather scarce. We show that amphioxus (*Branchiostoma japonicum*) ADAT2 (BjADAT2) contains the active site 'HxE-PCxxC' and the key residues for target-base-binding, and amphioxus ADAT3 (BjADAT3) harbors both the N-terminal positively charged region and the C-terminal pseudo-catalytic domain important for recognition of substrates. The sequencing of BjADAT2-transformed *Escherichia coli* genome suggests that BjADAT2 has the potential to target *E. coli* DNA and can deaminate at T<u>C</u>G and G<u>A</u>A sites in the *E. coli* genome. Biochemical analyses further demonstrate that BjADAT2, in complex with BjADAT3, can perform A-to-I editing of tRNA and convert C-to-U and A-to-I deamination of DNA. We also show that BjADAT2 preferentially deaminates adenosines and cytidines in the loop of DNA hairpin structures of substrates, and BjADAT3 also affects the type of DNA substrate targeted by BjADAT2. Finally, we find that C89, N113, C148 and Y156 play critical roles in the DNA editing activity of BjADAT2. Collectively, our study indicates that BjADAT2/3 is the sole naturally occurring deaminase with both tRNA and DNA editing capacity identified so far in Metazoa.

[1] Institute of Evolution & Marine Biodiversity and Department of Marine Biology, Ocean University of China, 266003 Qingdao, China. [2] Institute of Evolution & Marine Biodiversity, KLMME, Ocean University of China, 266003 Qingdao, China. [3] Laboratory for Marine Biology and Biotechnology, Laoshan Laboratory, 266237 Qingdao, China. [4] These authors contributed equally: Zhan Gao, Wanyue Jiang, Yu Zhang. ✉email: gaozhan@ouc.edu.cn; sczhang@ouc.edu.cn

The tRNA adenosine deaminase (TadA), first identified in yeast[1], is an enzyme that catalyzes the deamination of adenosine (A) to inosine (I) at the 'wobble' position of the anticodon of particular tRNAs, an essential process for degenerate codon decoding during translation[2,3]. The bacterial TadA is considered to form a homodimer, while in eukaryotic organisms, TadA, also called ADAT (adenosine deaminase acting on tRNA), is a heterodimer formed by the association of ADAT2, the catalytic subunit, and ADAT3, the non-catalytic subunit which serves only a structural role[2]. Both TadA and ADAT2/3 all contain an active site motif 'HxE-PCxxC', in which 'H' and 'C' residues are responsible for zinc ion coordination and 'E' residue for proton shuttling in deamination, except that the essential 'E' is missing in ADAT3, with its place taken by a catalytically inactive residue such as valine in most eukaryotic ADAT3s[4].

A-to-I deamination in the anticodon of tRNAs mediated by TadA or ADAT2/3 heterodimer has been documented for more than two decades[2,3,5], but the study on their DNA editing activity remains limited. Currently, only trypanosome ADAT2/3 heterodimer has been shown to perform cytidine-to-uridine (C-to-U) but not A-to-I deamination of DNA[6]. TadA is unable to perform A-to-I deamination of DNA, though its structure is closely similar to those of several DNA editing enzymes of the APOBEC family[7,8], and the mutagenesis of Asp108 to Asn in TadA of E. coli (EcTadA) enables it to convert A to I on DNA substrates[9]. Therefore, to our best knowledge, no report hereto shows the presence of a naturally occurring tRNA-editing deaminase that is capable of A-to-I deamination of DNA. In this study, we identify and characterize the adenosine-to-inosine tRNA-editing deaminase from amphioxus. Remarkably, the recombinant enzyme consisting of two subunits (BjADAT2 and BjADAT3) can not only perform A-to-I editing of tRNA but also C-to-U and A-to-I deamination of DNA. To our best knowledge, this is the sole naturally occurring deaminase with both tRNA and DNA editing capacities reported in Metazoa.

## Results

**Molecular characteristics of BjADAT2 and BjADAT3.** We identified two genes (GenBank accession numbers: MZ560714 and MZ560713) encoding ADAT2 and ADAT3 in the amphioxus B. japonicum, termed BjADAT2 and BjADAT3, respectively. BjADAT2 consisted of 208 residues, which shared 30 to 47% identity to the ADAT2 of eukaryotes, including mammals, fishes, shellfishes, and protozoa, and about 30% identity to prokaryotic ortholog TadA[4] (Supplementary Fig. 1). BjADAT3 comprised 385 residues, which was 19–40% identical to the ADAT3 of eukaryotes ranging from protozoa to mammals (Supplementary Fig. 2). Molecular modeling revealed that BjADAT2, like all known ADAT2 and TadA, contained a core five-stranded β sheet structural element surrounded by α helices, as well as the active site 'HxE-PCxxC' within the potential substrate-binding pocket formed by the loop 1, loop 3, loop 5, loop 7, and C-terminal helix (Fig. 1a).

Multiple sequence alignment revealed that the key residues for target-base-binding were all conserved in eukaryotic ADAT2 including BjADAT2 (i.e., V29, N45, H56 and A57) as well as in prokaryotic TadA (Fig. 1b). It was notable that the key residues for recognition of the 5′ flanking nucleoside in Mus musculus ADAT2 were Q127, N128 and V164, corresponding to D103, D104 and S138 in Staphylococcus aureus TadA, and the key residues for recognition of the 3′ flanking nucleoside in M. musculus ADAT2 were I40, V62 and N63, corresponding to G22, R44 and E45 in S. aureus TadA. These changes were thought to confer broader substrate specificity to ADAT2 than TadA[10]. We found that in BjADAT2, the residues for recognition of 5′

flanking nucleoside were P112, N113 and A149, and the residues for recognition of 3′ flanking nucleoside were G25, V47 and N48; these were closely similar to those of M. musculus ADAT2, but substantially different from those of S. aureus TadA. It was also notable that the residue D108 in E. coli TadA (equivalent to D104 in S. aureus TadA) was shown to be a determinant for targeting RNA or DNA because an alteration of this D to N enabled EcTadA to target DNA substrates[9,11]. We uncovered that a conserved N existed at the position equivalent to D108 of E. coli TadA in all known eukaryotic ADAT2 proteins, including BjADAT2, suggesting the presence of potential DNA editing activity in ADAT2.

Sequence comparison showed that BjADAT3, like all known ADAT3, harbored a positively charged (rich of K and R) region in the N-terminus, and a pseudo-catalytic domain (the catalytic residue 'E' in TadA/ADAT2 is replaced by a 'V') in the C-terminus (Supplementary Fig. 2). As both the N-terminal positively charged region and the C-terminal pseudo-catalytic domain in yeast and mouse ADAT3 were shown indispensable for the selective recognition of substrates[10], BjADAT3 may thus be potentially involved in substrate selection. The data above prompted us eagerly to investigate if BjADAT2/BjADAT3 complex can catalyze A-to-I deamination in DNA in the following experiments.

**BjADAT2 induced mutagenesis in E. coli.** At first, we tested the DNA editing activity of BjADAT2 and BjADAT3 through an E. coli-colony formation assay that is routinely used for screening DNA deaminases[12]. We found that expression of BjADAT2 in E. coli significantly increased bacterial rifampicin resistance (Rif^R) mutation frequency (fourfold) relative to the empty vector control, whereas expression of BjADAT3 had little effect on the Rif^R frequency (Fig. 2a). Compared with the known DNA deaminases, such as M. musculus AID (MmAID)[13] and ABE8e-derived TadA8e[14], the mutagenic activity of BjADAT2 revealed by Rif^R mutation frequency was about 2.5-fold and 2-fold weaker than that of MmAID and TadA8e, respectively (Fig. 2a). The mutagenic activity of BjADAT2 was also assessed by the mutagenic assay using nalidixic acid (Nal) instead of Rif. It showed that BjADAT2 induced a ninefold increase in the number of Nal^R mutant colonies over the empty vector group, though this increased level was lower than that of MmAID (23-fold increase). In sharp contrast, overexpression of EcTadA had no effects on mutation frequency in E. coli, consistent with its inability to target DNA[9]. Importantly, we found that the mutagenic activity of BjADAT2 depended on the presence of the 'HxE-PCxxC' motif characteristic of deaminases because BjADAT2-RQGG mutant (mutations in four residues H56R, E58Q, C92G and C95G) was unable to induce mutation in E. coli. These indicated that BjADAT2 was capable of inducing mutagenesis in E. coli.

To determine the target substrate of BjADAT2, we analyzed the DNA mutational spectrum by sequencing the rpoB gene in several independent Rif^R colonies. In E. coli, expression of cytidine deaminase was shown to generate C:G to T:A transition mutations[12], while expression of adenosine deaminase was shown to generate A:T to G:C transition mutations[9]. Compared with the empty vector transformants, the BjADAT2 transformants generated more C:G to T:A mutations (1.15-fold) and A:T to G:C mutations (2.25-fold), suggesting that BjADAT2 could deaminate both cytidine and adenosine on genome DNA (Fig. 2b, c). Notably, there was a difference in mutation distribution within the rpoB gene between the BjADAT2 transformants and controls: the C:G to T:A transitions at positions S512 (tct) and S531 (tcc) and the A:T to G:C transitions at positions L511 (ctg), S512 (tct) and L533 (ctc) in the BjADAT2 transformants were not seen in

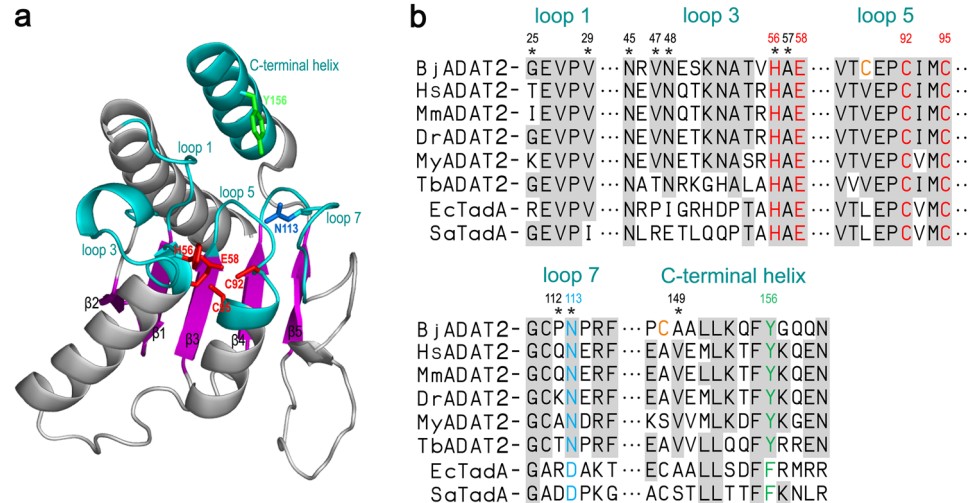

**Fig. 1 Structure and multiple sequence alignment of BjADAT2. a** The catalytic core structure of BjADAT2 (residues 4–160). BjADAT2 contains a core five-stranded β sheet (purple) structural element surrounded by α helices. The potential active sites (H56, E58, C92 and C95) are shown as red sticks, and the surrounding loops and helices are colored in cyan. The residues N113 (equivalent to EcTadA D108 that was shown to be a determinant for targeting RNA or DNA) and Y156 (equivalent to EcTadA F149 that was important for the deamination activity) are shown as sticks. **b** Multiple sequence alignments of ADAT2 from *Branchiostoma japonicum, Homo sapiens, Mus musculus, Danio rerio, Mizuhopecten yessoensis* and *Trypanosoma brucei*, as well as their prokaryotic ortholog TadA from *Escherichia coli* and *Staphylococcus aureus* (Supplementary Fig. 1). The putative recognition sites for the target base (V29, N45, H56 and A57), the 5' flanking nucleoside (P112, N113 and A149) and the 3' flanking nucleoside (G25, V47 and N48) are indicated by asterisks. The residues C89 and C148 of BjADAT2 are indicated in orange font.

any of the controls; however, the transition positions of the controls showed sign of reduced mutation in the BjADAT2 transformants (Fig. 2b).

The cytidine deaminase activity of BjADAT2 was also supported by examining the spectrum of *gyrA* gene mutations in the Nal[R] colonies, which revealed that 73% of the mutations in the BjADAT2 transformants were C:G to T:A, contrasting to 40% in the empty vector transformants (Fig. 2d). However, the A:T to G:C proportion in *gyrA* mutations was nearly the same between the BjADAT2 and control groups. This was possibly because the number of Nal[R] selectable sites is small, and the target adenosine residing in the local sequences context is unfavorable for BjADAT2, thus leading to low adenosine deaminase activity.

To confirm the adenosine deaminase activity of BjADAT2, we analyzed the *rpoB* gene mutant for Rif[R] in an *E. coli* strain lacking *alkA* gene, which encodes an alky-adenine DNA glycosylase with hypoxanthine excision activity, thus involving in mutation avoidance following deamination[15,16]. BjADAT2 expression in the *alkA*− background yielded an A:T to G:C proportion (31%) that was greater than in the *alkA*+ background (18%) (Supplementary Fig. 3). By contrast, no significant difference was observed in the A:T to G:C proportion between *alkA*− and *alkA*+ background (both were 8%) in the empty vector groups, well agreeing with the previous reports that deleting *alkA* had little effect on the rate of spontaneous base substitutions in *E. coli*[17,18]. These data demonstrated that BjADAT2 could induce A:T to G:C transition by deaminating adenosine in DNA.

**BjADAT2 displayed preferential deamination at TCG and GAA sites in the *E. coli* genome.** The above analysis of the *rpoB* mutations revealed that 74% of BjADAT2-induced C:G to T:A mutations occurred in the context of TC:GA (mutable positions underlined), much higher than that (25%) of the control group (Fig. 2b), suggesting that BjADAT2 might prefer targeting the 5'-TC context. For the *gyrA* mutations, as the potential cytidine deamination sites for Nal[R] were all in the TC:GA context, we found that BjADAT2 mediated higher Nal[R] mutation frequency (ninefold higher than that of control), compared with relatively

lower Rif[R] mutation frequency (4.8-fold higher than that of control) (Fig. 2a). It was clear that BjADAT2 showed a conspicuous TC preference, which was also reported in the non-canonical cytidine deamination events mediated by the adenine base editors (ABEs)[19,20].

Despite the evidence that BjADAT2-induced mutation was non-random and presumably dependent on local sequence context, we could not unbiasedly determine the sequence preference for adenosine and cytidine deamination with these datasets because there were only a very limited number of sites in the target gene that could lead to the selected phenotypes. Thus, we further performed the Cirseq method[21] for detecting genome-wide rare mutations of the BjADAT2-, BjADAT2-RQGG-, and empty vector transformants, based on which we analyzed BjADAT2-associated mutational patterns (sequence context of mutation sites). As expected, the frequency of C:G to T:A mutation in the group expressing BjADAT2 is significantly higher than the empty vector control group and the inactive BjADAT2-RQGG mutant control group (Fig. 2e), further confirming the cytidine deamination activity of BjADAT2. The neighbor bases of these cytidine mutation sites are biased to upstream T at the 5' flank and downstream G at the 3' flank (Fig. 2f), supporting that 5'TCG3' was a mutational target motif of BjADAT2. For the A:T to G:C mutation frequency, although the difference in total number was not significant compared with the control groups, the BjADAT2 group had more mutations occurring at 5'GAA3' context (Fig. 2g), suggesting that GAA was also a target motif of BjADAT2. In addition to these two types of mutations described above, the other four types of mutational spectrum (A:T to T:A, G:C to T:A, A:T to C:G, and G:C to C:G) between each group showed no statistical difference (Fig. 2e). These data together indicated that BjADAT2 preferentially deaminates TCG and GAA sites in the *E. coli* genome.

**BjADAT2 preferred deamination of adenosine and cytidine in DNA hairpins in vitro.** Recombinant BjADAT2 and BjADAT3 were individually expressed in *E. coli* and purified, and their enzyme activities were tested by in vitro tRNA deamination assay

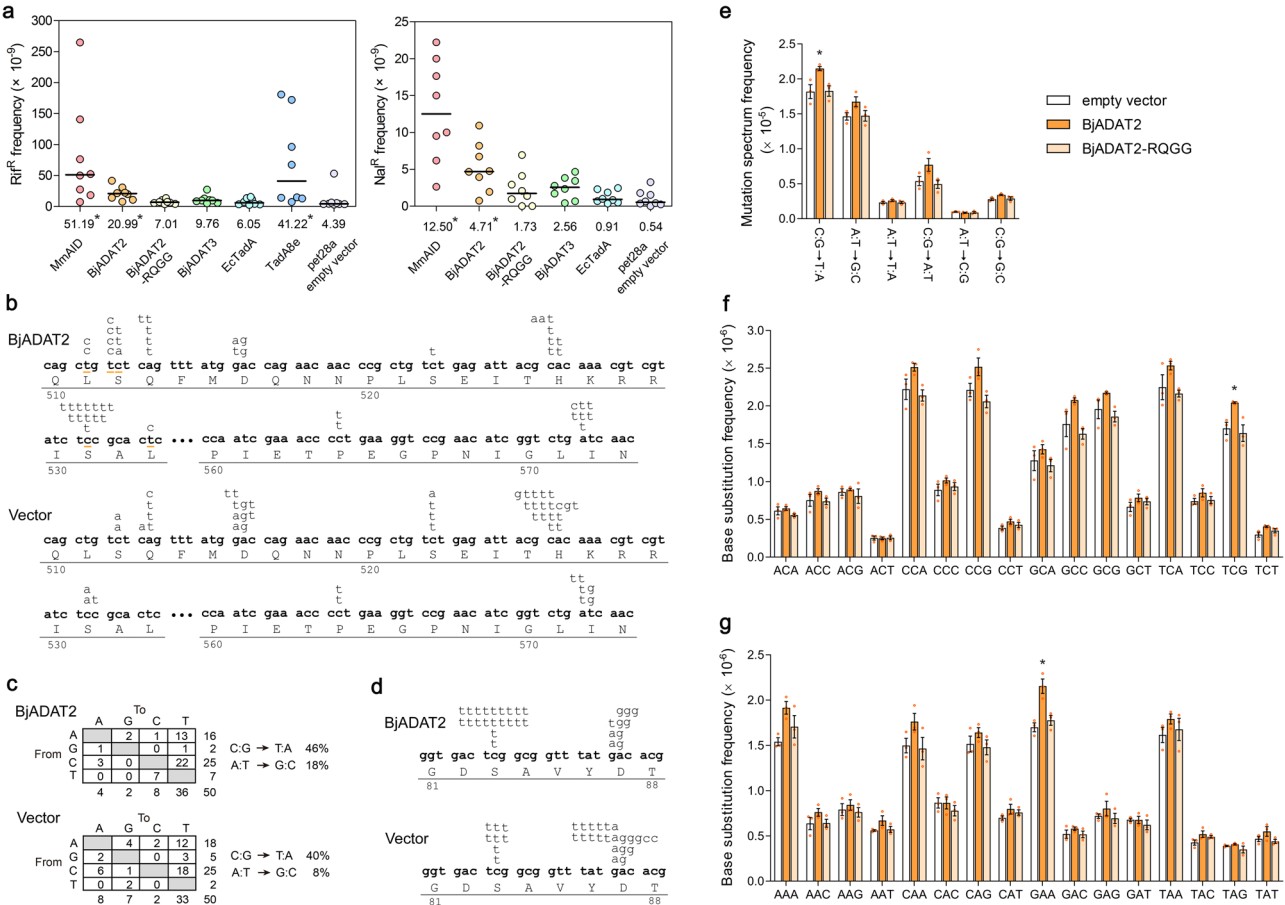

**Fig. 2 DNA editing activities of BjADAT2 in *E. coli*. a** Mutation frequencies for Rif[R] or Nal[R] in *E. coli* transformed with MmAID, BjADAT2, BjADAT2-RQGG (an active-site mutant), BjADAT3, EcTadA, TadA8e, or empty pet28a vector. Each point represents the mutation frequency of an independent culture. The horizontal bars represent the median (numerical value below each column), and statistical differences of each deaminase compared to the empty vector were assessed using a two-tailed Wilcoxon rank-sum for unpaired data, *$p < 0.05$. **b** Comparison of the distribution of independent *rpoB* mutations identified in Rif[R] colonies obtained from BjADAT2- and empty vector transformants (50 independent cultures for each group and one clone per culture). The sequence shown here contains all of the mutations (shown above the DNA sequence) detected in this assay. Each clone contains one and only one of the mutations. Compared with the control group, the unique sites of C:G to T:A and A:T to G:C transitions in the BjADAT2 group were underlined in orange. **c** Comparison of the pattern of *rpoB* nucleotide substitutions obtained from BjADAT2- and empty vector transformants. **d** Comparison of the distribution of independent *gyrA* mutations identified in Nal[R] colonies obtained from BjADAT2- and empty vector transformants (30 independent cultures for each group and one clone per culture). The sequence shown here contains all of the mutations detected in this assay. Each clone contains one and only one of the mutations. **e** Base-substitution mutation rates in *E. coli* Transetta strains transformed with pet28a-BjADAT2, pet28a-BjADAT2-RQGG or empty vector (three independent cultures for each group). Mutation spectrum of each transformant with IPTG-induction at 28 °C for 24 h. The mutation rates of either C:G to T:A substitutions (**f**) or A:T to G:C substitutions (**g**) occurring on the focal base of all possible 16 contexts were analyzed, respectively. We combined nucleotide triplets that are reverse complemented; for example, 5'TGT3' is taken in the same context as 5'ACA3', and 5'TTT3' is taken in the same context as 5'AAA3'. Data are represented as the mean ± SEM from three independent bacterial cultures per group. The statistical differences between the BjADAT2 group and the control (empty vector and BjADAT2-RQGG) groups were assessed using one-way ANOVA followed by Tukey's post-test, *$p < 0.05$.

(Supplementary Fig. 4). Neither BjADAT2 nor BjADAT3 showed any A-to-I editing activity, but BjADAT2, in complex with BjADAT3, could perform A-to-I editing of amphioxus tRNA$^{Val}_{(AAC)}$, and the apparent first-order deamination rate constant ($k_{app}$) of tRNA deamination was $0.0287 ± 0.00219$ min$^{-1}$. Because the optimal RNA substrate for ADAT2 was the adenosines on the anticodon loop structure of tRNA, we wondered if BjADAT2 could mediate DNA deamination in a structure-specific fashion. Thus, we compared the deamination efficiency of BjADAT2 on DNA hairpin structure substrates (i.e., hpDNA-A and hpDNA-C, containing a single adenosine or cytidine in the loop region, respectively) and single-stranded linear structure substrates (i.e., ssDNA-A and ssDNA-C, containing a single adenosine or cytidine in the substrates, respectively). As shown in Fig. 3, both adenosine

deamination of hpDNA-A and cytidine deamination of hpDNA-C were clearly observed under BjADAT2 treatment, and the change of the hairpin substrates to linear substrates resulted in about a threefold decrease in adenosine- and cytidine-deamination ratios. By contrast, no product band was seen in the lane of hpDNA-G or hpDNA-T treated with BjADAT2. Moreover, the BjADAT2-E58A protein purified in the same way as BjADAT2 showed no deamination activity (Supplementary Fig. 5f, g), ruling out the possibility that the observed deamination activity of BjADAT2 arose from a contaminant in the recombinant protein samples. In addition, BjADAT2-mediated adenosine- and cytidine deamination was inhibited by deoxycoformycin (DCF), an adenosine deaminase specific inhibitor, or tetrahydrouridine (THU), a cytidine deaminase specific inhibitor, or 1,10-*o*-phenanthroline, a zinc chelator. These

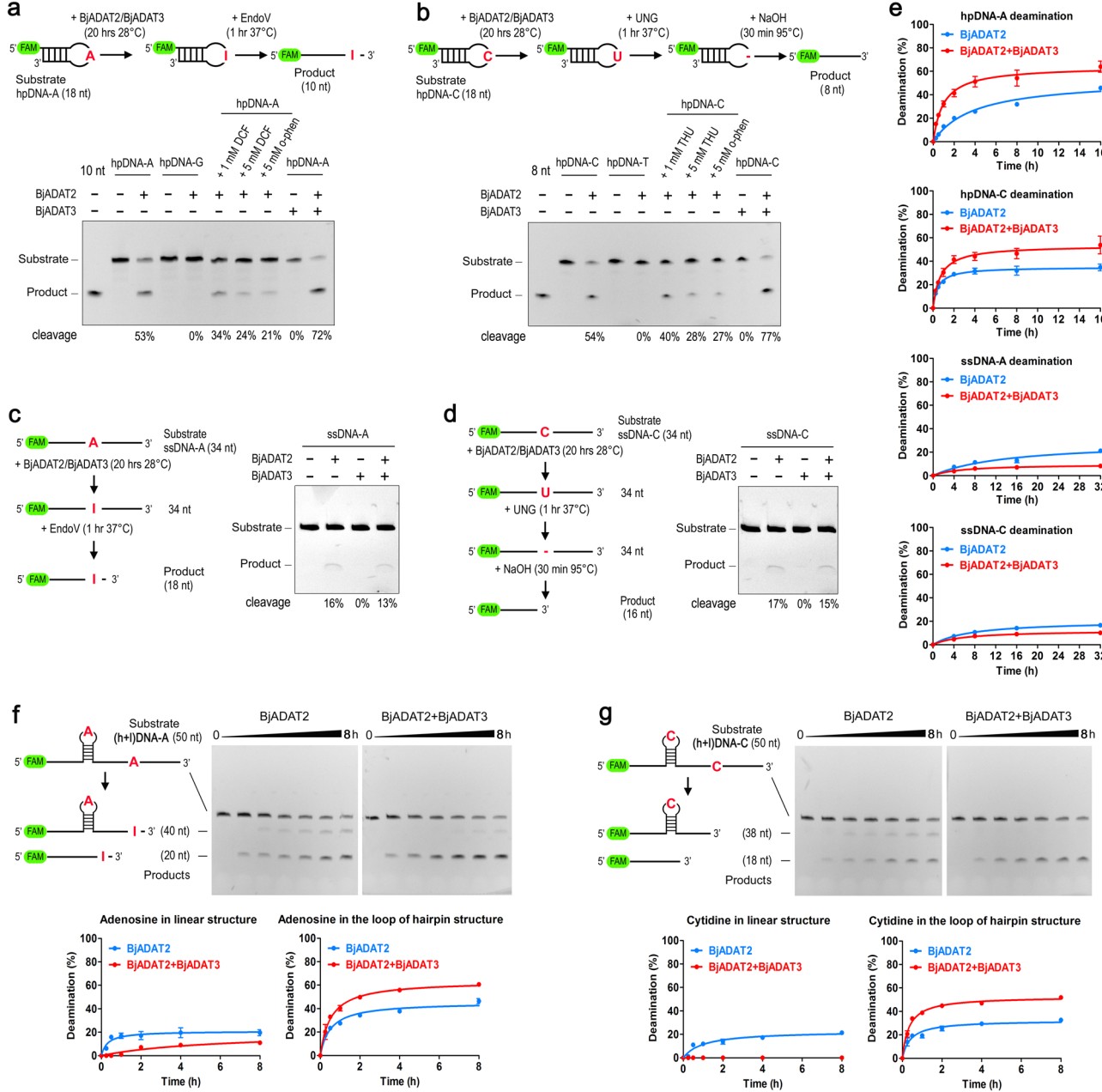

**Fig. 3 DNA deamination activity of BjADAT2 and BjADAT3. a** Schematic and representative TBE-Urea gel of the adenosine deamination assay with a hairpin DNA substrate hpDNA-A. Substrate hpDNA-G was used as a control. FAM 6-carboxyfluorescein, DCF deoxycoformycin, o-phen 1,10-*o*-phenanthroline. **b** Schematic and representative TBE-Urea gel of the cytidine deamination assay with a hairpin DNA substrate hpDNA-C. Substrate hpDNA-T was used as a control. THU tetrahydrouridine. **c** Schematic and representative TBE-Urea gel of the adenosine deamination assay with a linear substrate ssDNA-A. **d** Schematic and representative TBE-Urea gel of the cytidine deamination assay with a linear substrate ssDNA-C. **e** Kinetic of BjADAT2 (with or without BjADAT3) measured using hpDNA-A, hpDNA-C, ssDNA-A, and ssDNA-C. The TBE-Urea gels are shown in Supplementary Fig. 5. The fraction of deaminated DNA was plotted as a function of time and fit to a single exponential equation to extract $k_{app}$. **f, g** Kinetic analysis of (h+l) DNA-A and (h+l)DNA-C deamination by BjADAT2 or BjADAT2/3 complex. Raw images of the representative TBE-Urea gels were shown in (**f, g**). Data are represented as the mean ± SEM from three independent experiments.

indicated that BjADAT2 deaminated adenosine and cytidine in a zinc-dependent manner.

The $k_{app}$ of BjADAT2-induced hpDNA-A deamination ($0.00206 \pm 0.00019$ min$^{-1}$) was comparable with that of ABE7.10-induced DNA adenosine deamination ($0.0010 \pm 0.00030$ min$^{-1}$)[22]. We compared the kinetics of BjADAT2-induced DNA deamination of the hairpin substrates and the linear substrates. The $k_{app}$ was sixfold higher for the hpDNA-A substrate than for the ssDNA-A substrate ($0.00032 \pm 0.00004$ min$^{-1}$). Similarly, the $k_{app}$ was 24-fold

higher for hpDNA-C ($0.00801 \pm 0.00037$ min$^{-1}$) than for ssDNA-C ($0.00033 \pm 0.000026$ min$^{-1}$) (Fig. 3e). These data together indicated that BjADAT2 preferentially deaminates adenosines and cytidines in the hairpin loop of substrates.

**BjADAT3 affected the type of substrate DNA bound by BjADAT2.** It was known that ADAT3, as a non-catalytic subunit, served a structural role in the adenosine deamination of tRNA[10,23]. To date, only trypanosome ADAT2 was shown to be

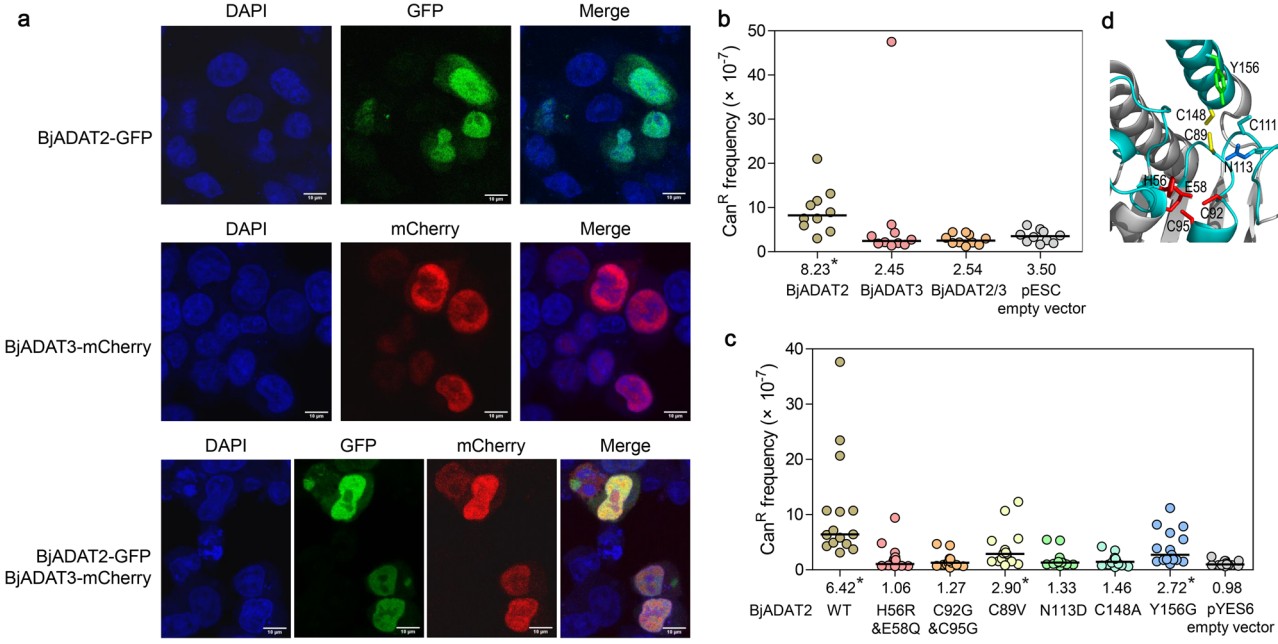

**Fig. 4 DNA editing activity of BjADAT2/BjADAT3 in the eukaryotic cell nucleus. a** Subcellular localization of BjADAT2 and BjADAT3 in HEK293T cells. pcDNA3.1-BjADAT2-GFP and pcDNA3.1-BjADAT3-mCherry were individually or co-transfected into HEK293T cells. The nucleus was stained by DAPI. The cells were imaged by a confocal microscope. The size of the scale bar is 10 μm. **b** Mutagenic activities of BjADAT2 and BjADAT3 in *S. cerevisiae*. Mutation frequencies for Can$^R$ in *S. cerevisiae* transformed with pESC-BjADAT2, -BjADAT3, or -BjADAT2/3 vectors, or empty vector. **c** Mutation frequencies for Can$^R$ in *S. cerevisiae* transformed with pYES6-BjADAT2-wide type, pYES6-BjADAT2 mutants including H56R&E58Q, C92G&C95G, C89V, N113D, C148A and Y156G (indicated in (**d**)), or empty vector. Each point represents the mutation frequency of an independent culture. The horizontal bars represent the median (numerical value below each column), and statistical differences of each recombinant vector compared to the empty vector were assessed using a two-tailed Wilcoxon rank-sum for unpaired data, *$p < 0.05$.

able to deaminate cytosines of DNA in the presence of ADAT3 in vitro[6], but the specific role of the ADAT3 subunit in this DNA editing reaction was unclear. We demonstrated here that BjA-DAT3, though lacking catalytic function, could enhance the deamination activity of BjADAT2 toward hairpin structure substrates, including hpDNA-A and hpDNA-C (Fig. 3a, b). This was also supported by the results of kinetic analysis (Fig. 3e), which showed that the $k_{app}$ of hpDNA-A deamination of BjADAT2/BjADAT3 complex $(0.01092 \pm 0.00062 \, \text{min}^{-1})$ was 5.3-fold higher than that of BjADAT2 alone, and the $k_{app}$ of hpDNA-C deamination by the complex $(0.01066 \pm 0.00083 \, \text{min}^{-1})$ 1.3-fold higher than that of BjADAT2 alone. On the contrary, the $k_{app}$ of ssDNA-A deamination of BjADAT2/BjADAT3 complex $(0.00016 \pm 0.000034 \, \text{min}^{-1})$ was lower than that of BjADAT2 alone, and the $k_{app}$ of ssDNA-C deamination by the complex $(0.00020 \pm 0.000025 \, \text{min}^{-1})$ lower than that of BjADAT2 alone. Furthermore, when we used longer single-stranded DNA molecules that have both hairpin and linear deamination sites, i.e., (h+l)DNA-A and (h+l)DNA-C as the substrates, the results showed that BjADAT3 limited the linear sites that can be deaminated by BjADAT2 but enhanced deamination of hairpin sites (Fig. 3f, g). These data suggested that BjADAT3 affects the DNA substrate types that BjADAT2 binds, including enhancement of the deamination of hairpin substrates and blockage of the deamination of linear substrates.

**BjADAT2/BjADAT3 performed DNA editing in the eukaryotic cell nucleus.** HEK293T cells were employed to examine the subcellular location of BjADAT2 and BjADAT3 (because no amphioxus cell lines are available now). We found that BjADAT2 was mainly localized in the nucleus and in a small portion in the cytoplasm, whereas BjADAT3 was preferentially localized in the

nucleus (Fig. 4a). Given that ADAT2 and ADAT3 were expected to work together as a heterodimer, we thus co-expressed BjA-DAT2 and BjADAT3 in the cells and found that they indeed co-localized in the nucleus, suggesting the possibility of BjADAT2 and BjADAT3 to form heterodimer and function in the nucleus.

Next, the yeast-based mutation assay was performed by testing the canavanine-resistant (Can$^R$) mutation frequencies of *S. cerevisiae* and sequencing the *can1* gene from Can$^R$ colonies to test the mutagenic activity of BjADAT2/BjADAT3 in eukaryotic cells. It was found that BjADAT2 expression induced an increase in Can$^R$ mutations in *S. cerevisiae*, but BjADAT3 expression did not. Interestingly, co-expression of BjADAT2 and BjADAT3 in *S. cerevisiae* resulted in a decrease of the Can$^R$ mutation frequency to the level of the empty vector control group (Fig. 4b). Possibly, it might be caused by the limitation of BjADAT3 to BjADAT2 on the utilization of target substrates. To verify this assumption, we analyzed the DNA mutation sites in the *can1* gene and found that, in the co-expression group, 75% of the deamination sites (i.e., C:G to T:A and A:T to G:C mutation sites) were located on hairpin loop structure. This ratio was higher than that of both the BjADAT2 group (57%) and the empty vector control group (0%) (Supplementary Table 1). That is to say, although BjADAT3 could promote BjADAT2 acting on the hairpin substrate, its major function in the BjADAT2-induced *can1* gene mutations was to block BjADAT2 acting on the linear substrates. Altogether, these data suggested that BjADAT2 and BjADAT3 could act jointly on DNA editing in the nucleus of eukaryotic cells.

Previous studies have shown that alteration of D108 to N enabled EcTadA to perform adenosine deamination on DNA substrates[9], and further alteration of F149 to Y was contributable to the stabilization of DNA substrates in a loop-like conformation[14]. In BjADAT2, the corresponding residues were N113 and Y156. We also examined the roles of the residues N113

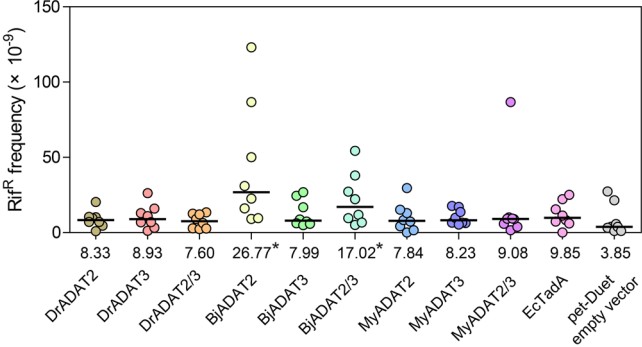

**Fig. 5 Mutagenic activities of fish ADAT2/3 and scallop ADAT2/3.** Mutation frequencies for Rif[R] in *E. coli* transformed with the petDuet vectors carrying the *ADAT2* and *ADAT3* genes (individually or in pairs) from *D. rerio*, *B. japonicum*, and *M. yessoensis*, or with EcTadA or empty petDuet vector. Each point represents the mutation frequency of an independent culture. The horizontal bars represent the median (numerical value below each column), and statistical differences of each deaminase compared to empty petDuet vector were assessed using a two-tailed Wilcoxon rank-sum for unpaired data, *$p < 0.05$. These genes were all correctly expressed in *E. coli* (Supplementary Fig. 9).

and Y156 in the BjADAT2-induced DNA editing process. We generated two BjADAT2 mutants with alteration of N113 to D and Y156 to G and determined their DNA editing capacity via the yeast Can[R] model system. As shown in Fig. 4c, the BjADAT2 mutant with N113D, resembling the active-site mutants (BjADAT2 mutant with H56R and E58Q and BjADAT2 mutant with C92G and C95G), completely lost DNA editing capacity. The BjADAT2 mutant with Y156G was also found to have about a twofold decrease in the Can[R] ratio compared with wild-type BjADAT2. In addition, we found that the residues C89 (usually being a valine in the ADAT2 from other eukaryotes) in loop 5 and C148 (usually being an alanine or serine in the ADAT2 from other eukaryotes) in the C-terminal helix of BjADAT2 were unique among all eukaryotic ADAT2s (Fig. 1b). Thus, we generated C89V mutant and C148A mutant, and determined their DNA editing capacity. The C89V mutant lost about half of the DNA editing activity of wild-type BjADAT2, while the C148A mutation completely abolished the DNA editing capacity (Fig. 4c). These data indicated that the residues C89, N113, C148 and Y156 played important roles in DNA editing activity of BjADAT2.

**DNA editing activity was sporadically distributed in eukaryotic ADAT2/3.** As BjADAT2/BjADAT3 exhibited DNA editing activity, and more than half of the substrate-binding residues, especially N113, were observed to be conserved in eukaryotic ADAT2 proteins (Fig. 1b), we thus wondered if the DNA editing activity was widely distributed in eukaryotic ADAT2/3. We cloned ADAT2 and ADAT3 genes from the invertebrate *Mizuhopecten yessoensis* (MyADAT2/3) and the vertebrate *Danio rerio* (DrADAT2/3) and evaluated their mutagenic activity by the colony formation assay as described above: the ADAT2 and ADAT3 genes were expressed individually or in pairs in *E. coli*, and then the Rif[R] colony counting was recorded. It was found that neither expression of each subunit of MyADAT2/3 and DrADAT2/3 nor co-expression of MyADAT2/3 and DrADAT2/3 induced an increase in the Rif[R] mutation frequency. By contrast, both the expression of BjADAT2 and co-expression of BjADAT2/3 in *E. coli* caused a significant increase in the Rif[R] mutation frequency relative to the empty vector control, though the expression of BjADAT3 showed little influence on the Rif[R] mutation frequency (Fig. 5). These suggested that the ADAT2/3 with catalyzing DNA

mutation activity might be sporadically distributed across the tree of life.

## Discussion

It is known to date that only protozoa trypanosome ADAT2/3 heterodimer can perform C-to-U but not A-to-I deamination of DNA[6], and mutated EcTadA with Asp108 to Asn can convert A to I on DNA substrates[9]. No reports suggest the existence of a tRNA-editing deaminase with a naturally occurring capacity of A-to-I deamination of DNA. In this study, we clearly demonstrate that the BjADAT2/3 complex of primitive chordate amphioxus not only performs A-to-I editing of tRNA but also C-to-U and A-to-I deamination of DNA, providing an example of a naturally occurring deaminase with tRNA and DNA editing capacity in Metazoa.

Currently known natural DNA deaminases in eukaryotes include two types: cytidine deaminase and adenosine deaminase. The former mainly refers to the AID/APOBEC family, including AID and APOBEC1-4 in vertebrates and various AID/APOBEC-like deaminases in invertebrates[24,25]. They usually target cytidines on ssDNA substrates[26]. By contrast, only two deaminases with DNA adenine editing activity have been discovered, i.e., RNA-specific adenosine deaminases (ADARs) 1 and 2, both of them targeting mismatched adenosines on the DNA strand of DNA/RNA hybrids[27]. Several deaminases also edit RNA, such as APOBEC1 (which targets C-to-U on ssRNA[28] and has been applied in both RNA- and DNA-base editors[29−32]) and ADAR1 and 2 (which target A-to-I on dsRNA[33] and have been applied in RNA base editor[34,35]). Nevertheless, all the deaminases above have remarkable structural homology and highly conserved catalytic HxE and PCxxC motifs[4], indicating an evolutionary relationship among them. DNA deaminases are thought to be originated from bacterial toxin systems, followed by extensive diversification into multiple eukaryotic clades and rapid evolution under the drive of arms-race with viruses and genomic retroelements[4,24]. In this study, a natural DNA deaminase, BjADAT2/3, that can deaminate both cytidines and adenosines on ssDNA substrates is demonstrated, representing a special version standing out from the broader array of deaminases pitted in such arms races.

Given that the active sites 'HxE-PCxxC' and key residues for substrate binding, e.g., N113 and Y156 in BjADAT2 observed in amphioxus and protozoa trypanosome ADAT2/3 enzymes are all highly conserved in ADAT2 orthologs, we thus wonder if the DNA editing activity is widely distributed in eukaryotic ADAT2/3. We tested our hypothesis with the ADAT2/3 genes from one invertebrate (scallop) and one vertebrate (zebrafish). Surprisingly, neither scallop ADAT2/3 nor zebrafish ADAT2/3 show DNA editing activity. Previous studies also reported that the yeast ADAT2/3 complex expressed in *E. coli* did not exhibit mutagenic activity[36], and the human ADAT2 (fused with Cas9 D10A nickase) was unable to perform A to I deamination in HEK293T cells[9]. These data suggest the possibility that the ADAT2/3 with DNA editing activity occurs sporadically across the tree of life. It is known that the specificity of a deaminase could be radically altered by changing a single amino acid[9,37]. We examined the amino acid sequences of the substrate-binding pocket of ADAT2 orthologs and found that the residues C89 and C148 of BjADAT2 were unique among all available orthologs. Mutation of C89V and C148A significantly reduced the DNA editing activity of BjADAT2, suggesting that C89 and C148 are pivotal residues contributing to BjADAT2 activity. Interestingly, the residue C89 in BjADAT2 is equivalent to L84 in EcTadA, whose side chain protrudes toward the DNA backbone and therefore affects the substrate option of TadA[9,20]. Moreover, C89,

C148, and another cysteine residue, C111 (equivalent to EcTadA A106, mutation at this residue affecting DNA-binding activity of ABEs[9,20]), are found closely located in BjADAT2 (Fig. 4d). We speculate that the three cysteines may form a Zn-ion binding site, to stabilize the structure of enzyme or to facilitate the substrate binding to the pocket. Similarly, additional Zn-binding sites were also observed in the substrate-binding regions of several AID/APOBEC-like deaminases, implying their auxiliary role in the enzyme-substrate interaction[24]. More experimental evidence is needed to support this hypothesis, and this will shed more light on the evolution of deaminases.

The sequencing of BjADAT2-transformed *E. coli* genomes reveals that the triplet context preferences of BjADAT2 are 5'G$\underline{A}$A3' for adenosine deamination and 5'T$\underline{C}$G3' for cytidine deamination. This difference in the preferred contexts is also observed in the modified TadA catalytic domains from various versions of adenine base editors, such as ABE7.10, ABEmax and ABE8e, all of which exhibit broad context compatibility for the target adenine base and a context restriction for the target cytosine base (the preferred context is 5'T$\underline{C}$N3', N = A, T, C or G)[19,20]. In the TadA domain of these ABEs, the target base and 5' flanking base are flipped out into the substrate-binding pocket, away from the U-shaped ssDNA backbone, whereas the 3' flanking base shows an altered conformation. These flanking bases make contact with the residues from the pocket[14,20]. As cytosine is smaller than adenine, and the catalytic site is located deep inside the substrate-binding pocket, thus a shift of the DNA backbone toward the rim of the pocket is needed to allow the pyrimidine ring of cytosine base to reach the same position as that of the hexagonal ring of the adenine base in the structure[20]. This shift of the DNA backbone may result in the fact that the optimal flanking base-binding manner for the target adenine base is not suited for the target cytosine base. Recently, the TadA-derived editors have been developed very quickly through directed evolution and rational mutagenesis approaches[14,38–40], and some of the latest versions enable both adenine and cytosine editing[41–44]. Also, an evolved RNA editing enzyme from the ADAR2 deaminase domain has both A-to-I and C-to-U editing activity on dsRNA substrates[45]. Further structural studies will elucidate the difference in the mechanism of adenine-substrate and cytosine-substrate engagement by these deaminases.

Our biochemical analysis shows that hairpin DNA sites are optimal substrates for BjADAT2, probably because the enforced sharp turn of a hairpin loop facilitates the target base and flanking base to flip out. In addition, BjADAT2 may preferentially bind the substrate with an anticodon stem-loop-like structure, as the enzyme contains a string of positively charged amino acids in the C-terminus (Supplementary Fig. 1), which is supposed to be crucial for binding the phosphate backbone of tRNA[46]. We also show that the deamination of hairpin DNA substrates by BjADAT2 is enhanced by BjADAT3. By contrast, for the deamination of the linear DNA substrate, BjADAT3 exhibits an inhibitory effect, as evidenced by the deamination activity assay in vitro and the mutagenic activity assay in yeast cells. This bifunctional property of BjADAT3 may be due to the classical role of ADAT3, which uses the N-terminal positively charged region to bind the stem region of the tRNA substrate[10].

Although the DNA editing activity of BjADAT2 was weaker than that of AID, the mutagenic activity of BjADAT2 in the eukaryotic nucleus was confirmed by the yeast-colony formation assay. Imaginably it would be a destabilizing factor for amphioxus if BjADAT2 was allowed to edit its genome arbitrarily. We uncover that BjADAT3 efficiently limits the mutagenic activity of BjADAT2 in yeast cells by blocking BjADAT2 acting on the linear DNA substrates. Even so, a tight regulation mechanism of this enzyme's activity in the nucleus needs further study.

An example of modulating ADAT2/3-mediated mutagenesis is that the mutagenicity of trypanosome ADAT2/3 (TbADAT2/3) can be restricted by the methyltransferase TRM140a[6,47]. As TRM140a could increase the binding affinity of TbADAT2/3 for tRNA, the complex may be sequestered away on the tRNA, precluding interaction and potentially rampant deamination of the genome by TbADAT2/3[48]. Regulation of ADAT2/3 activity through interaction with other proteins or cofactors should be an important mechanism in the cellular control of ADAT2/3-mediated mutagenesis.

It has been demonstrated that amphioxus is one of the most genetically diverse animals sequenced to date[49–51]. Somatic mutation frequency per nucleotide in amphioxus ($10^{-3}$)[52] is orders of magnitude higher than in other multicellular eukaryotes, including mammals ($10^{-7}$)[53,54], fly ($10^{-5}$)[55] and coral ($10^{-7}$)[56], while mutational processes in amphioxus are unknown. We reanalyzed the data of amphioxus somatic mutations reported previously[52], and the results showed that C:G to T:A and A:T to G:C are dominant substitutions with the frequency of $8.3 \times 10^{-3}$ and $6.6 \times 10^{-3}$, respectively (Supplementary Fig. 6), implying a high frequency of cytidine and adenosine deamination on genomic DNA. It would be of great interest to further study the relevance between BjADAT2/3 and amphioxus somatic mutations. In addition, the mutagenic activity may confer BjADAT2/3 another important function, like vertebrate APOBECs acting as part of the defense against a wide variety of viruses[57,58] or like invertebrate AID/APOBEC-like deaminases taking part in the defense against bacteria[25]. In the amphioxus genome, no AID/APOBEC homologous genes have been identified thus far[24], but BjADAT2/3 had evolved a specific ability to mutate ssDNA, especially in hairpin loops, which may have the potential to aid in the defense against ssDNA viruses with intricately folded structures. Therefore, the study on the immune function of BjADAT2/3 is currently underway.

## Methods

**Animals and cells.** Adult amphioxus *B. japonicum* collected from the sandy bottom of the sea near Qingdao, China, were cultured in aerated seawater at room temperature and fed with single-celled algae once a day. Zebrafish (*D. rerio*) aged 3 months and adult Japanese scallops (*M. yessoensis*) were purchased from Nanshan market in Qingdao. Mice (*M. musculus*) aged 11 months were purchased from Jinan Pengyue Laboratory Animal Breeding Co., Ltd. All animals were maintained following the ethical guidelines, and all experimental protocols were approved by the Institutional Animal Care and Use Committee of the Ocean University of China. HEK293T cell line was a gift from Dr. Jianfeng Zhou, Ocean University of China. The HEK293T cells were cultured at 37 °C in DMEM medium (Hyclone) supplemented with 10% (v/v) fetal bovine serum (Gibco). *E. coli* Transetta (DE3) strain was purchased from TransGen Biotech Co., Ltd., and its *alkA::accC1* (Gm$^R$) derivative was constructed by Sangon Biotech Co., Ltd.

**Gene cloning and sequence analysis.** Total RNAs extracted from adult *B. japonicum* were used to synthesize cDNAs with a reverse transcription kit (TransGen Biotech, China, #AT311) according to the manufacturer's instructions. Partial cDNA fragments of BjADAT2 and BjADAT3 were amplified by PCR using ExTaq DNA polymerase (TaKaRa, #RR001) and the primer pairs that were designed on the basis of the hypothetic ADAT2 and ADAT3 mRNA sequences of *B. floridae* in NCBI (accession numbers: XM_035803301.1 and XM_035831516.1). The 5' and 3' RACE were performed using the SMARTer RACE 5'/3' Kit (Clontech, #634858) to obtain the complete cDNA sequence. The clones obtained were sequenced, and the overlapping sequences were assembled. All of the primers used in the PCRs hereafter were listed in Supplementary Table 2.

The assembled cDNA was analyzed for coding probability with the EditSeq in the DNASTAR software package (DNAStar, Madison, WI, USA). Homology searches in the GenBank database were carried out by the BLASTP network server (https://blast.ncbi.nlm.nih.gov/Blast.cgi) at the NCBI, and multiple alignments of the protein sequences generated using ESPript 3.x software (http://espript.ibcp.fr/ESPript/cgi-bin/ESPript.cgi)[59]. The three-dimensional (3D) structure of the catalytic core (residues 4–160) of BjADAT2 was predicted using the iterative threading assembly refinement (I-TASSER) program (https://zhanglab.ccmb.med.umich.edu/I-TASSER/)[60]. The quality of the 3D model was assigned a TM-score of $0.88 \pm 0.07$ by the I-TASSER program (a TM-score >0.5 indicates a model of

correct topology) and also evaluated by QMEANDisCo[61] and ProQ3D[62] servers. The QMEANDisCo Global score was 0.79 ± 0.07, and the ProQ3D value was 0.739.

**Construction of prokaryotic expression vectors.** The open reading frames (ORFs) of *BjADAT2* and *BjADAT3* were individually ligated into the EcoR I/Xho I sites of pet28a (Supplementary Table 2). The plasmids constructed were verified by sequencing and designated pet28a-BjADAT2 and pet28a-BjADAT3 (expressing N-terminal 6×His-tagged BjADAT2 and BjADAT3, respectively). The expression vectors for the inactive mutants of BjADAT2, including BjADAT2-RQGG (with mutations in four residues H56R, E58Q, C92G and C95G) and BjADAT2-E58A, were generated by site-directed mutagenesis from pet28a-BjADAT2 plasmid using Mut Express II Fast Mutagenesis Kit (Vazyme, China, #C214-01). To construct the expression vectors carrying the genes of known deaminases for control, PCR was performed to amplify the ORF of *M. musculus* AID (MmAID) from mouse spleen cDNAs[13] and *E. coli* TadA (EcTadA) from *E. coli* genome. The DNA fragment of the TadA8e domain was obtained by cloning the TadA7.10 domain from plasmid pCMV-ABE7.10, followed by site-directed mutation according to the sequence of the TadA8e domain of ABE8e[22]. The cloned fragments (containing the original stop codons at their 3′ ends) were inserted into the pet28a vector (Supplementary Table 2).

**Mutagenic activity assay in *E. coli*.** The mutagenic activity of BjADAT2 and BjADAT3 was tested through an *E. coli*-colony formation assay according to the method described by Petersen-Mahrt et al.[12]. In brief, *E. coli* strain transformed with the pet28a-BjADAT2 or pet28a-BjADAT3 vectors were grown in LB medium supplemented with kanamycin (50 μg/ml) and isopropyl β-D-1-thiogalactopyranoside (IPTG, 0.5 mM) at 28 °C for 24 h. Culture expressing MmAID was used as a positive control, those expressing EcTadA and catalytically inactive BjADAT2-RQGG mutant as negative controls, and that carrying pet28 empty vector as a blank control. The expression of these recombinant proteins was verified by Western blot using a mouse anti-His-tag monoclonal antibody (1:5000, CWBIO, China, #CW0286) as the primary antibody and horseradish peroxidase (HRP)-conjugated goat anti-mouse IgG antibody (1:8000, CWBIO, #CW0102) as the secondary antibody (Supplementary Fig. 7). Eight independent bacterial cultures per vector group were used for the following Rif^R- and Nal^R-colony formation assays. Rifampicin resistance (Rif^R) and nalidixic acid resistance (Nal^R) colonies were selected on LB agar plates with 100 μg/ml rifampicin or 40 μg/ml nalidixic acid, respectively. At the same time, aliquots of each culture at appropriate concentrations were plated onto LB agar plates in order to determine the number of viable cells in the cultures. Colonies were counted, and the mutation frequencies were determined as the ratio of the number of Rif^R or Nal^R colonies to the number of viable cells from the same culture. In addition, TadA8e was used in the Rif^R-colony formation assay to assess the relative editing activity of BjADAT2.

The identity of Rif^R mutation in *rpoB* gene and Nal^R mutation in *gyrA* gene were determined according to the method described by Petersen-Mahrt et al.[12]. In brief, 80 independent pet28a-BjADAT2- or pet28a empty vector-transformed colonies were grown in LB medium supplemented with kanamycin (50 μg/ml) and IPTG (0.5 mM) at 28 °C for 24 h. For each vector group, 50 cultures were plated onto LB (Rif^+) agar plates, and the other 30 cultures were plated onto LB (Nal^+) agar plates. A single colony from each plate was tested in a colony-PCR using KOD-Plus DNA polymerase (TOYOBO, #KOD-201) and the primer pair rpoB-S/AS (for Rif^R colonies) or gyrA-S/AS (for Nal^R colonies). PCR products were purified using Gel Extraction Kit (OMEGA, #D2500) and sequenced using the primer rpoB-S or gyrA-S, respectively. In addition, Rif^R mutation in the *rpoB* gene of the Transetta *alkA^−* strain was analyzed using the same method.

**E. coli Cirseq and rare-mutation analysis.** The *E. coli* Transetta strains transformed with the pet28a-BjADAT2, pet28a-BjADAT2-RQGG or pet28a empty vector were grown in LB medium supplemented with kanamycin (50 μg/ml) and IPTG (0.5 mM) at 28 °C for 24 h (three independent cultures per vector group). The genomic DNA was then extracted from each culture using MasterPure™ Complete DNA&RNA Purification Kit (Lucigen, #MC85200). We selected the sizeable fragments (90–120 bp) using Gel Extraction Kit (OMEGA, #D2500) after digesting the genomic DNA with NEBNext dsDNA fragmentase (NEB, #M0348S). The recovered fragments were denatured to single-strand linear DNA (ssDNA), and the ssDNA was ligated into circular DNA, which was used as a template for rolling circle amplification (RCA) with phi29 DNA polymerase[21]. The TruePrep DNA Library Prep Kit V2 for Illumina (Vazyme, #TD501) was used to construct libraries with the RCA products for Cirseq. Gel-based size selection was also performed after the library preparation to select fragments in the range of 500 bp to 750 bp, with an average insert size of ~300 bp. Sequencing was done using an Illumina Novaseq6000 sequencer with PE150. After trimming library adapters and removing reads with quality scores <20 using fastp v1.0[63], we analyzed the clean data with Python scripts modified from Acevedo et al.[64]. All statistics were done in R 3.6.3.

The mutation spectrum (frequency distribution of each mutation type) was calculated by the ratio of the number of observed base substitutions to the number of the corresponding nucleotide sites analyzed. For example, C:G to T:A mutation frequency was calculated by the ratio of the total number of C to T plus G to A mutations to the total number of C plus G sites analyzed. In addition, the C:G to T:A or A:T to G:C mutation at a certain context was calculated similarly, which means that the numerator was the number of mutations in both the 5′XC/AX3′ context and its reverse complement context, while the denominator was the number of C plus G sites or A plus T sites analyzed, respectively.

**Expression and purification of BjADAT2 and BjADAT3.** The *E. coli* Transetta strains transformed with the plasmid pet28a-BjADAT2 or pet28a-BjADAT3 were grown in LB medium (containing 50 μg/ml kanamycin) at 37 °C until the OD600 reached about 0.8. Subsequently, IPTG (0.1 mM) was added to the cultures in order to induce the expression of recombinant proteins. After further incubation at 22 °C for 24 h, the bacterial cells were harvested and sonicated in the lysis buffer (20 mM Tris-HCl pH 8.0, 100 mM KCl and 10 mM imidazole). The lysates were centrifuged at $12000 \times g$ at 4 °C for 20 min, and the supernatants collected were loaded onto a Ni-NTA resin column (GE Healthcare). The column was successively washed with imidazole gradient (10–50 mM) and then eluted with lysis buffer supplemented with 250 mM imidazole. The purified BjADAT2 and BjADAT3 were dialyzed to a buffer (pH 8.0) containing 20 mM Tris-HCl, 100 mM KCl, 1 mM MgCl₂, 1 mM dithiothreitol (DTT) and 10% glycerol. The N-terminal 6×His-tag of the recombinant proteins was not removed because it had no effects on the mutagenic activity of BjADAT2 or BjADAT3 (Supplementary Fig. 7). In addition, BjADAT2-E58A mutant was expressed and purified by the same methods. The purity of the recombinant proteins was analyzed by a 12% SDS-PAGE. The protein concentrations were determined by the BCA method.

**in vitro tRNA deamination assay.** Amphioxus tRNA^Val(AAC) sequence was searched from the GtRNAdb database (http://gtrnadb.ucsc.edu/GtRNAdb2/index.html), and the corresponding single-strand DNA was synthesized by Shanghai Sangon Biotech Co., Ltd. The synthesized DNA was amplified by PCR using KOD-Plus DNA polymerase (TOYOBO, #KOD-201) and the primer pair tRNA-S/AS (Supplementary Table 2). The PCR products were digested by Apa I and Nde I, followed by ligation into the Apa I/Nde I sites of the pGEM-T vector. The constructed pGEM-T-tRNA^Val(AAC) plasmid was used for in vitro transcription of tRNA^Val(AAC) using T7 RNA polymerase according to standard protocols. The transcribed tRNA was purified using phenol-chloroform extraction and then precipitated by NaAc precipitation. The tRNA precipitated was dissolved in DEPC-treated water, heated at 70 °C for 3 min, and then allowed to cool to room temperature. For the tRNA deamination assay, 1 μg tRNA was incubated with 1 μM BjADAT2 and/or 1 μM BjADAT3 in a final volume of 100 μl (reaction buffer: 20 mM Tris-HCl pH 8.0, 100 mM KCl, 1 mM MgCl₂ and 1 mM DTT in DEPC-treated water) at 25 °C for 2 h. The tRNA in the reaction buffer without BjADAT2 or BjADAT3 was used as the negative control. After incubation, the tRNAs were purified by the phenol-chloroform method and used for synthesizing cDNA by a reverse transcription kit (TransGen Biotech, #AT311) using tRNA-AS as the reverse transcription PCR primer. The synthesized cDNA was used for PCR amplification with KOD-Plus DNA polymerase (TOYOBO, #KOD-201) and the primer pair tRNA-S/AS. The PCR products were purified using Gel Extraction Kit (OMEGA, #D2500) and sequenced using the primer tRNA-AS. In addition, the tRNA deamination rate was analyzed as follows. 5′FAM-labeled tRNA^Val(AAC) was synthesized by Shanghai Sangon Biotech Co., Ltd. The tRNA substrate (0.36 μM) was incubated with BjADAT2 (1 μM) plus BjADAT3 (1 μM) for various times (0, 0.2, 0.5, 1, 1.5, 2 and 2.5 h) and then heated at 95 °C for 5 min. Subsequently, the heated mixture was added with 5U of *E. coli* Endonuclease V (EndoV; NEB, #M0305S) and NEBuffer 4 (NEB), incubated at 37 °C for 60 min, and then subjected to a 14% TBE-Urea gel electrophoresis, followed by imaging with a Mini-Chemi system (Sagecreation, China).

**in vitro DNA deamination assay.** The 6-FAM-labeled DNA substrates were synthesized by Shanghai Sangon Biotech Co., Ltd. The hairpin structure substrates included hpDNA-A, hpDNA-C, hpDNA-G and hpDNA-T [6-FAM-5′-GTCTGCTT(A/C/G/T)GTTTGCAGA-3′, in which the third base of the loop region (underlined) was A, C, G or T, respectively]. The single-stranded linear structure substrates included ssDNA-A and ssDNA-C [6-FAM-5′-(GGTT)₄(A/C)G(GGTT)₄-3′, in which the 17th base was A or C, respectively]. The substrates containing both hairpin and linear structures included (h+l)DNA-A and (h+l)DNA-C [6-FAM-5′-GT₁₀CTGCTTT(A/C)GTTGCAGAT₁₁(A/C)GT₁₀-3′]. The substrates with the hairpin structure were self-annealed by heating to 95 °C and snap-cooling.

For cytidine deamination assay, 1 μM BjADAT2 (with or without 1 μM BjADAT3) was incubated with 0.2 μM substrate ssDNA-C, hpDNA-C, hpDNA-T (negative control for hpDNA-C) or (h+l)DNA-C at 28 °C in deamination buffer containing 20 mM Tris-HCl pH 8.0, 100 mM KCl, 1 mM MgCl₂ and 1 mM DTT. Aliquots of 20 μl were removed at various time points (as indicated in figure legends) and heated at 95 °C for 10 min. Then, 5U of *E. coli* Uracil-DNA glycosylase (UDG; NEB, #M0280S) and UDG buffer (NEB) was added for a 60-min incubation period at 37 °C in order to excise the uracil. After uracil excision, NaOH was added to a final concentration of 150 mM and heated at 95 °C for 30 min in order to cleave the alkali-labile abasic site. Samples were run on a 14% TBE-Urea gel and imaged with a MiniChemi system (Sagecreation, China). The intensities of the uncleaved and cleaved DNA were analyzed using ImageJ software (National Institutes of Health). The apparent first-order deamination rate constant ($k_{app}$) was

calculated by GraphPad Prism 5. In addition, the deamination ratio of hpDNA-C by BjADAT2 was also tested in the presence of 5 mM tetrahydrouridine (THU, a cytidine deaminase inhibitor; MedChemExpress) or 1,10-o-phenanthroline (a zinc chelator; MedChemExpress).

For adenosine deamination assay, 1 μM BjADAT2 (with or without 1 μM BjADAT3) was incubated with 0.2 μM substrate ssDNA-A, hpDNA-A, hpDNA-G (negative control for hpDNA-A) or (h+l)DNA-A at 28 °C in deamination buffer containing 20 mM Tris-HCl pH 8.0, 100 mM KCl, 1 mM MgCl$_2$ and 1 mM DTT. Aliquots of 20 μl were removed at various time points (as indicated in figure legends) and heated at 95 °C for 10 min. Then, 5U of E. coli EndoV (NEB, #M0305S) and NEBuffer 4 (NEB) were added and incubated at 37 °C for 60 min in order to cleave the second DNA phosphodiester backbone 3′ to deoxyinosine[65]. Samples were run on a 14% TBE-Urea gel and imaged with a MiniChemi system (Sagecreation, China). The intensities of the uncleaved and cleaved DNA were analyzed using ImageJ software. The $k_{app}$ was calculated by GraphPad Prism 5. In addition, the deamination ratio of hpDNA-A by BjADAT2 was also tested in the presence of 5 mM deoxycoformycin (DCF, an adenosine deaminase inhibitor; MedChemExpress) or 1,10-o-phenanthroline (MedChemExpress).

**Subcellular localization**. In the subcellular localization assay, pcDNA3.1-BjA-DAT2-GFP and pcDNA3.1-BjADAT3-mCherry vectors were constructed (Supplementary Table 2). The constructed vectors were transfected individually or co-transfected into HEK293T cells using Lipofectamine 2000 Reagent (Invitrogen). Twenty-four hours after transfection, the cells were washed with PBS, fixed with 4% paraformaldehyde, and stained with 1 mg/ml DAPI. The samples were observed under a Leica TCS-SP8 confocal microscope.

**Mutagenic activity assay in S. cerevisiae**. To test the mutagenic activity of BjADAT2, BjADAT3 and BjADAT2/3 complex in Saccharomyces cerevisiae, canavanine resistant (Can$^R$)-colony formation assay was performed according to the method of Hoopes et al.[66]. The expression vectors pESC-BjADAT2, pESC-BjADAT3, pESC-BjADAT2-BjADAT3, as well as their tagged versions (Flag-tagged BjADAT2 and Myc-tagged BjADAT3), were constructed (Supplementary Table 2). The yeast strain BY4741 transformed with the expression vectors (carrying a his3 gene) was cultured respectively, to saturation in SC-His (synthetic complete medium without histidine) liquid medium containing 2% glucose. After being washed with sterile water, the cells were cultured in the SC-His liquid medium containing 2% galactose and 1% raffinose at 28 °C for 3 days. Culture carrying pESC empty vector served as a negative control. The expression of Flag-tagged BjADAT2 and Myc-tagged BjADAT3 were verified by Western blot using a mouse anti-Flag-tag monoclonal antibody (1:3000, Beyotime, China, #AF519) and a mouse anti-Myc-tag monoclonal antibody (1:3000, Beyotime, #AM926) as the primary antibody, respectively, and a HRP-conjugated goat anti-mouse IgG antibody (1:8000, CWBIO, #CW0102) as the secondary antibody for both of them (Supplementary Fig. 8). Note that the following Can$^R$-colony formation assay was performed using the untagged versions of BjADAT2 and BjADAT3, and 10 independent cultures per expression vector group were tested. Can$^R$ colonies were selected on SC-Arg (synthetic complete medium without arginine) agar plate with 60 μg/ml canavanine. At the same time, aliquots of each culture at appropriate concentrations were plated onto SC agar plates in order to determine the number of viable cells in the cultures. Colonies were counted, and the mutation frequencies were determined as the ratio of the number of Can$^R$ colonies to the number of viable cells from the same culture.

Next, a single Can$^R$ colony from each plate of the pESC-BjADAT2-, pESC-BjADAT2-BjADAT3-, or empty vector group was used to identify Can$^R$ mutation in the can1 gene according to the method of Kozmin et al.[67]. In brief, the selected colonies were grown in SC-Arg (Can$^+$) liquid medium to saturation. The genomic DNA of each culture was extracted by Yeast DNA Kit (OMEGA, #D3370) and used for PCR amplification of the can1 gene with KOD-Plus DNA polymerase (TOYOBO, #KOD-201) and the primer pair CAN-S/AS. The PCR products were purified using Gel Extraction Kit (OMEGA, #D2500) and sequenced using three primers: CAN-P1, CAN-P2, and CAN-AS (Supplementary Table 2). The hairpin structure around the mutation site was predicted by UNAFold Web Server (http://www.unafold.org/).

**Mutagenic activity of BjADAT2 mutants**. To investigate the key residues for the DNA editing activity of BjADAT2, yeast expression vector pYES6-BjADAT2 and its six mutants (i.e., H56R&E58Q, C92G&C95G, C89V, N113D, C148A and Y156G) were constructed (expressed proteins had no tags; Supplementary Table 2) and used for the following yeast Can$^R$-colony formation assay. BY4741 transformed with these vectors were separately grown to saturation in YPD liquid medium supplemented with blasticidin (50 μg/ml), washed in sterile water, and then grown in the blasticidin selective liquid medium containing 2% galactose and 1% raffinose (instead of glucose) at 28 °C for 3 days. Culture carrying pYES6 empty vector was used as a negative control. The Can$^R$ mutation frequencies were determined using 15 independent yeast cultures per expression vector group. The expression of target proteins in yeast was evaluated using BY4741 transformed with pYES6-BjADAT2-His and its mutant versions (Supplementary Table 2 and Supplementary Fig. 8b).

**Assay for the mutagenic activity of scallop and fish ADAT2/3 s**. PCR was performed to amplify the ORFs of M. yessoensis ADAT2 and ADAT3 genes (GenBank accession numbers: XM_021517453.1 and XM_021483739.1) and D. rerio ADAT2 and ADAT3 genes (GenBank accession numbers: XM_005160618.4 and NM_001005300.3) (Supplementary Table 2). We used the dual expression vector petDuet for expressing MyADAT2/3, DrADAT2/3 and BjADAT2/3 individually or in pairs. These expression vectors were transformed into E. coli Transetta strains, and the transformants grown in LB medium supplemented with 100 μg/ml ampicillin and 0.5 mM IPTG at 28 °C for 24 h. Culture carrying petDuet empty vector was used as a negative control. The expression of His-tagged ADAT2s and S-tagged ADAT3s was verified by Western blot using a mouse anti-His-tag monoclonal antibody (1:5000, CWBIO, #CW0286) or a mouse anti-S-tag monoclonal antibody (1:5000, Sangon Biotech, China, #D191105) as the primary antibody, respectively, followed by an HRP-conjugated goat anti-mouse IgG antibody (1:8000, CWBIO, #CW0102) as the secondary antibody (Supplementary Fig. 9). The Rif$^R$-colony formation test was performed using eight independent cultures per expression vector group, and the Rif$^R$ mutation frequencies of each expression vector group were analyzed to evaluate their mutagenic activity.

**Statistics and reproducibility**. For most experiments, at least three independent biological replicates were performed. The statistical analysis was performed using GraphPad Prism (GraphPad Software, La Jolla, CA, USA). In the colony formation assays, statistical significance between the two groups was determined by a two-tailed Wilcoxon rank-sum for unpaired data. For comparing the mutation spectrum of multiple groups, statistical significance was determined by one-way ANOVA followed by Tukey's post-test.

**Reporting summary**. Further information on research design is available in the Nature Portfolio Reporting Summary linked to this article.

## Data availability

Raw reads of Cirseq were deposited in NCBI Sequence Read Archive with BioProject no. PRJNA753282. Nucleic acid sequence information of BjADAT2 and BjADAT3 was deposited in GenBank with the following accession codes: MZ560714 and MZ560713. Source data for figures can be found in Supplementary Data 1. All other data are available upon a reasonable request from the corresponding author.

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

## Acknowledgements

This work was supported by the grants of the National Natural Science Foundation of China (32070514 to Z.G., 31961123002 to H.L.); and the Young Taishan Scholars Program of Shandong Province (tsqn201812024 to H.L.). We appreciate the computing resources provided by IEMB-1, a high-performance computation cluster operated by the Institute of Evolution and Marine Biodiversity.

## Author contributions

Z.G. and S.Z. conceived and coordinated the project; Z.G., H.L. and S.Z. designed the experiments; Z.G., W.J., Y.Z., L.Z., M.Y., H.W., Z.M., B.Q., and X.J. performed experiments; Z.G. and W.J. performed the statistical analysis; Z.G. and S.Z. wrote the paper, with input from the other authors. All authors reviewed the manuscript and approved the final version.

## Competing interests

The authors declare no competing interests.

**Additional information**

