## [Peer Review File · Communications Biology]

Reviewers' comments:

Reviewer #1 (Remarks to the Author):

Adenosine deaminases that act on tRNA (ADATs) perform a key function by deaminating the tRNA anticodon loop and allowing for wobble base pairing. While in bacteria this is the function of a single enzyme (TadA), in eukaryotes this functionality is performed by a heterodimer of two proteins that have cytidine deaminase folds. This prompted Alfonzo and colleagues, years ago, to look for cytidine deaminase functionality in these enzymes as well, which they found in DNA (but not tRNA).

In this manuscript, Zhang and colleagues describe an ADAR2/3 homolog from amphioxus, that can catalyze A to I editing in tRNA. The ADAT2 portion of this heterodimer however, can perform C-to-U editing in DNA (similarly to the ADAT2 described in *T. brucei*) but also, uniquely, A-to-I editing in DNA. ADAT3 enhances, but is not required for activity.

I have two comments:

1) experimentally, evidence of induction of mutation by deamination in the E. coli but also yeast systems is usually supported by repeating the experiment in the context of a repair knockout that cannot deal with the specific mutation (UNG in the case of C to U, or AAG/MPG in the case of A to I). For completion, it would be important for the authors to do this.

2) The finding is important both for evolutionary biology but also for synthetic biology (as enzymes that can perform both functions have not been shown to exist *in vivo*; an evolved ADAR enzyme that can catalyze C to U editing has been reported by F. Zhang and colleagues, and it will be interesting for these authors to discuss them comparatively.

Reviewer #2 (Remarks to the Author):

In this manuscript, Zhan Gao et al. investigated amphioxus Adenosine-to-inosine tRNA-editing enzyme 2 (BjADAT2) and showed that BjADAT2 has the potential to target *E. coli* DNA and can deaminate at TCG and GAA sites in the *E. coli* genome and preferentially deaminates adenosines and cytidines in the loop of DNA hairpin structures of substrates, and BjADAT3 also affects the type of DNA substrate targeted by BjADAT2. This manuscript is a well-done enzymological study and examines the characteristics of BjADART2 in detail.

However, this manuscript merely examines an enzyme in a previously unexamined animal species for ADAT2 that has already been identified, lacks impact. The authors studied DNA editing in eukaryotic cells of BjADAT2, but does it edit its own genomic DNA? If there is a possibility, the heterogeneity of the genomic DNA of the amphioxus should be proven. If there is no such possibility, then the meaning of the study in natural science is unclear. If the research is to utilize BjADAT2 as a genome editing tool, it is necessary to show its superiority compared to other tools (enzymes), but such research has not been done. Therefore, I recommend that the authors submit it to a more suitable journal.

Comments

1. The authors show the 3D structures of BjADAT2 in Fig. 1, how were these results obtained? Also, is there any reason to believe that these structures, especially the surface view on the right, is correct?

2. In Fig. 2, the authors show mutation frequencies and distributions, but how many clones were analyzed for each study?

3. In line 161 and 162, the authors suggested that BjADAT2 might prefer targeting the 5'-TC context while deaminating cytidine in DNAs. The results certainly suggest this, but how does it work for double-stranded DNAs, where the bases are inside of the sugar backbone and should form complementary base pairs with each other?

Reviewer #3 (Remarks to the Author):

Gao reported "Amphioxus adenosine-to-inosine tRNA-editing enzyme that can perform C-to-U and A-to-I deamination". DNA editing activity triggered by adenosine-to-inosine tRNA-editing enzyme has been harnessed as genome editing tool, while deaminase from nature possessing high DNA editing have not been identified. In the present study, authors tested BjADAT2 and BjADAT3, resulting in the identification of BjADAT2 possessing relative high DNA editing. In general, the study is well designed and well performed. While, there are several issues which should be addressed.

1. A positive control(i.e.,ecTadA from ABE8e) for the A to G editing activity should be included in at least one study, thus, we could assess the relative editing activity of BjADAT2.
2. In Fig 1b, it is better to add human ADAT2 alignment.
3. As to the study of BjADAT2 in E. coli and yeast, is there any recombination between endogenous ADAT2 and BjADAT2? Or BjADAT2/ADAT3 compound dimer?
4. tRNA editing activity of ADAT2 is as we expected, which may not be treated as a selling point in this study. Thus, this section could be minimized.
5. The biological function of BjADAT2/ADAT3 may be further discussed although so far it is not clear.

Reviewer #4 (Remarks to the Author):

Reviewer's comments:

The manuscript has been written so nicely and I really congratulate the authors for bringing a very nice topic which will contribute a lot to the scientific society.

Although some minor revisions are needed for the upgrading the quality of the manuscript. Recently some works have been published regarding the C to U and A to I editing which has really made a huge contribution and I think those citations are needed and at the same time at the discussion part some relevant sentences needed to be added.

Such as:

<https://doi.org/10.3390/genes13091636>

RNA editing of BFP, a point mutant of GFP, using artificial APOBEC1 deaminase to restore the genetic code | Scientific Reports (nature.com)

<https://doi.org/10.1093/protein/gzz005>

Moreover, the result section could be in a more detailed form and in more relevance with the discussion.

Finally, I would like to say that with those minor revisions this manuscript can be accepted in this reputed journal. I would like to revise the manuscript after the revision.

Reviewer #1:

Adenosine deaminases that act on tRNA (ADATs) perform a key function by deaminating the tRNA anticodon loop and allowing for wobble base pairing. While in bacteria this is the function of a single enzyme (TadA), in eukaryotes this functionality is performed by a heterodimer of two proteins that have cytidine deaminase folds. This prompted Alfonzo and colleagues, years ago, to look for cytidine deaminase functionality in these enzymes as well, which they found in DNA (but not tRNA).

In this manuscript, Zhang and colleagues describe an ADAR2/3 homolog from amphioxus, that can catalyze A to I editing in tRNA. The ADAT2 portion of this heterodimer however, can perform C-to-U editing in DNA (similarly to the ADAT2 described in *T. brucei*) but also, uniquely, A-to-I editing in DNA. ADAT3 enhances, but is not required for activity.

I have two comments:

1) experimentally, evidence of induction of mutation by deamination in the *E. coli* but also yeast systems is usually supported by repeating the experiment in the context of a repair knockout that cannot deal with the specific mutation (UNG in the case of C to U, or AAG/MPG in the case of A to I). For completion, it would be important for the authors to do this.

Response: Thanks for the suggestion. We have performed the analysis of *rpoB* mutation for Rif^R in an *E. coli* strain lacking *alkA* gene (*alkA* gene encodes an alky-adenine DNA glycosylase, i.e., AAG), and the result provided further evidence on the adenosine deaminase activity of BjADAT2 (the primary highlight of this manuscript). Please see the revised manuscript. For the experiment of cytidine deamination, our study here (including the mutational spectrum and distribution in the *rpoB* and *gyrA* genes) has demonstrated the cytidine deaminase activity of BjADAT2. In addition, to unbiasedly determine the deaminase activity, we analyzed the genome-wide mutations using a Cirseq method, and the results provided strong evidence that BjADAT2 could deaminate cytidine and adenosine in the *E. coli* genome. Therefore we did not re-do the experiment in the *ung*⁻ background.

2) The finding is important both for evolutionary biology but also for synthetic biology (as enzymes that can perform both functions have not been shown to exist in vivo; an evolved ADAR enzyme that can catalyze C to U editing has been reported by F. Zhang and colleagues, and it will be interesting for these authors to discuss them comparatively).

Response: Thanks. We have discussed the deaminases with both A to I and C to U editing activities comparatively. Please see the section of the revised manuscript.

Reviewer #2:

In this manuscript, Zhan Gao et al. investigated amphioxus Adenosine-to-inosine tRNA-editing enzyme 2 (BjADAT2) and showed that BjADAT2 has the potential to target *E. coli* DNA and can deaminate at TCG and GAA sites in the *E. coli* genome and preferentially deaminates adenosines and cytidines in the loop of DNA hairpin structures of substrates, and BjADAT3 also affects the type of DNA substrate targeted by BjADAT2. This manuscript is a well-done enzymological study and examines the characteristics of BjADART2 in detail.

However, this manuscript merely examines an enzyme in a previously unexamined animal species for ADAT2 that has already been identified, lacks impact. The authors studied DNA editing in eukaryotic cells of BjADAT2, but does it edit its own genomic DNA? If there is a possibility, the heterogeneity of the genomic DNA of the amphioxus should be proven. If there is no such possibility, then the meaning of the study in natural science is unclear. If the research is to utilize BjADAT2 as a genome editing tool, it is necessary to show its superiority compared to other tools (enzymes), but such research has not been done. Therefore, I recommend that the authors submit it to a more suitable journal.

Response: Thanks for the comments and suggestions. We have performed the experiments as suggested. We have analyzed the amphioxus somatic mutations and discussed the possible biology functions of BjADAT2/3. Please see the last paragraph of the Discussion section. In brief, somatic single-base substitution frequency in amphioxus (10^{-3}) is orders of magnitude higher than that of other multicellular eukaryotes ($10^{-5}\sim 10^{-7}$). Among the single-base substitutions, C:G to T:A and A:T to G:C are dominant types with the frequency of 8.3×10^{-3} and 6.6×10^{-3} , respectively (Supplementary Fig. S9), implying a high frequency of cytidine and adenosine deamination on amphioxus genomic DNA. We are currently investigating the mutational mechanism in amphioxus and its relevance with BjADAT2/3 in future.

Comments

1. The authors show the 3D structures of BjADAT2 in Fig. 1, how were these results obtained? Also, is there any reason to believe that these structures, especially the surface view on the right, is correct?

Response: The 3D structure model of BjADAT2 was predicted using I-TASSER program (Yang and Zhang, *Nucleic Acids Research*, 2015, 43:W174-181), followed by quality assessments using QMEANDisCo and ProQ3D servers, which confirmed that it was good in quality. The model here is to show that BjADAT2 contains a core five-stranded β sheet structural element surrounded by α helices, as well as the active site 'HxE-PCxxC' within the potential substrate-binding pocket formed by the loop 1, loop 3, loop 5, loop 7, and C-terminal helix. The information can be completely embodied

by the cartoon view in Fig. 1a. Given that the surface view on the right is just another form of the cartoon view, we have removed the surface view, and it has no effects on the conclusions of molecular characteristics and any other analyses.

2. In Fig. 2, the authors show mutation frequencies and distributions, but how many clones were analyzed for each study?

Response: Thanks. The description of the number of clones has been added in the figure legends. The mutagenic activity assay in *E. coli* (Fig. 2a-c) was performed according to the classical method (Petersen-Mahrt et al., Nature, 2002, 418:99-103). In Fig. 2a, each group contains eight independent cultures (each point represents the mutation frequency of an independent culture). In Fig. 2b and 2c, the data are collected from 50 independent cultures of BjADAT2-transformed *E. coli* and an equal number of controls (one clone per culture). Similarly, 30 independent cultures are collected for each group in Fig. 2d. In Fig. 2e-g, the data of each group are collected from three independent cultures.

3. In line 161 and 162, the authors suggested that BjADAT2 might prefer targeting the 5'-TC context while deaminating cytidine in DNAs. The results certainly suggest this, but how does it work for double-stranded DNAs, where the bases are inside of the sugar backbone and should form complementary base pairs with each other?

Response: We have modified this sentence. Our study demonstrated that the targeting of BjADAT2 is single-stranded DNAs. Previous studies also reported that many deaminases, e.g., AID and APOBECs, target single-stranded DNAs (Chaudhuri et al., Nature, 2003, 422:726-730; Salter et al., Trends in Biochemical Sciences, 2016, 41:578-594). We speculate that BjADAT2 can deaminate cytidine and adenosine when single-stranded DNAs are exposed, such as in the process of DNA transcription or replication. We appreciate your valuable comments, and will continue to study the mutagenic mechanism and physiological function of BjADAT2 in future.

Reviewer #3:

Gao reported "Amphioxus adenosine-to-inosine tRNA-editing enzyme that can perform C-to-U and A-to-I deamination". DNA editing activity triggered by adenosine-to-inosine tRNA-editing enzyme has been harnessed as genome editing tool, while deaminase from nature possessing high DNA editing have not been identified. In the present study, authors tested BjADAT2 and BjADAT3, resulting in the identification of BjADAT2 possessing relative high DNA editing. In general, the study is well designed and well performed. While, there are several issues which should be addressed.

1. A positive control (i.e., ecTadA from ABE8e) for the A to G editing activity should be included in at least one study, thus, we could assess the relative editing activity of BjADAT2.

Response: A good suggestion. The TadA8e (derived from ABE8e) has been added to the study in *E. coli*, and its mutagenic activity for Rif^R is 2-fold higher than that of BjADAT2.

2. In Fig 1b, it is better to add human ADAT2 alignment.

Response: We have added human ADAT2 in the alignment.

3. As to the study of BjADAT2 in *E. coli* and yeast, is there any recombination between endogenous ADAT2 and BjADAT2? Or BjADAT2/ADAT3 compound dimer?

Response: This is an interesting question. We checked the data of *E. coli* Cirseq in this study, and found that there was no recombination between endogenous TadA and BjADAT2, indicating that the enhancement of mutation frequency was indeed induced by wild-type BjADAT2. As to the study of BjADAT2 in yeast, we are sorry for not addressing the question above. We speculate that endogenous ADAT2 and ADAT3 have little effects on the mutagenic activity of BjADAT2 in yeast, because the increasement of Can^R mutation frequency in BjADAT2-transformed yeast (5-fold) is comparable to that of Rif^R in BjADAT2-transformed *E. coli* (4-fold). Also, the mutation frequency for Can^R of wild-type BjADAT2-transformant is significantly higher than that of any inactive BjADAT2-transformants. In addition, co-expression of BjADAT2 and BjADAT3 in yeast resulted in decrease of the Can^R mutation frequency to the background level, indicating the important role of BjADAT3 in the BjADAT2/3 dimer.

4. tRNA editing activity of ADAT2 is as we expected, which may not be treated as a selling point in this study. Thus, this section could be minimized.

Response: We have modified this section, including moving the result description and Fig. 3 to the Supplementary Information.

5. The biological function of BjADAT2/ADAT3 may be further discussed although so far it is not clear.

Response: Thanks for the suggestion. We have discussed the possible biological function of BjADAT2/3. Please see the last paragraph of the Discussion section.

Reviewer #4:

Reviewer's comments:

The manuscript has been written so nicely and I really congratulate the authors for bringing a very nice topic which will contribute a lot to the scientific society.

Although some minor revisions are needed for the upgrading the quality of the manuscript.

Recently some works have been published regarding the C to U and A to I editing which has really made a huge contribution and I think those citations are needed and at the same time at the discussion part some relevant sentences needed to be added.

Such as:

<https://doi.org/10.3390/genes13091636>

RNA editing of BFP, a point mutant of GFP, using artificial APOBEC1 deaminase to restore the genetic code | Scientific Reports (nature.com)

<https://doi.org/10.1093/protein/gzz005>

Moreover, the result section could be in a more detailed form and in more relevance with the discussion.

Finally, I would like to say that with those minor revisions this manuscript can be accepted in this reputed journal. I would like to revise the manuscript after the revision.

Response: Thanks for the comments. We have modified the section Discussion and cited these wonderful works.

If you have any question, please do not hesitate to contact us.

Yours sincerely,

Zhan Gao, PhD

Institute of Evolution & Marine Biodiversity and Department of Marine Biology

Ocean University of China

Qingdao 266003, China

E-mail: gaozhan@ouc.edu.cn

REVIEWERS' COMMENTS:

Reviewer #1 (Remarks to the Author):

The authors have been quite responsive to reviewer comments. This is a nice paper that offers an exciting addition to the literature of base editing enzymes.

Reviewer #2 (Remarks to the Author):

This manuscript by Gao et al. has been improved with revisions and is a great enzymology, but still lacks impact. As it stands, Gao et al. have isolated ADART2, which has already been reported, from a novel species of sluggish and have characterized it. It has certainly a novel enzyme activity, but it is not a major expansion of scientific knowledge and not expected to have a significant impact.

In their discussion, the authors note the relevance of this enzyme in somatic mutation data from Amazonia, where C:G to T:A and A:T to C:G mutations were predominant in amphibians. If that guess is correct, that would have a high impact, wouldn't it? Why don't the authors pursue that conjecture and get some evidence? As reviewer 1 pointed out, this would require at least introducing the gene into eukaryotic cells and confirming its genomic mutation, which does not seem to be a very difficult experiment. It would also be better to confirm DNA editing activity similar to this enzyme in other amphibians.

Additional Comments,

From line 337, the authors described "Several DNA deaminases also edit RNA, such as APOBECs and ADARs." However, as the name suggests, ADAR is an enzyme that catalyzes RNA and does not work on DNA. APOBECs were also originally discovered as an enzyme that edits apolipoprotein B mRNA in the intestine. The APOBEC family is diverse, with some enzymes catalyzing primarily DNA and others both DNA and RNA, but APOBEC1 is not believed to act on DNA. It should be corrected.

Reviewer #3 (Remarks to the Author):

The concerns have been well addressed.

We would like to express our wholehearted thanks to all the reviewers for their professional and valuable comments and suggestions.

Reviewer #1:

The authors have been quite responsive to reviewer comments. This is a nice paper that offers an exciting addition to the literature of base editing enzymes.

Response: Thanks for your affirmation.

Reviewer #2:

This manuscript by Gao et al. has been improved with revisions and is a great enzymology, but still lacks impact. As it stands, Gao et al. have isolated ADART2, which has already been reported, from a novel species of sluggish and have characterized it. It has certainly a novel enzyme activity, but it is not a major expansion of scientific knowledge and not expected to have a significant impact.

In their discussion, the authors note the relevance of this enzyme in somatic mutation data from Amazonia, where C:G to T:A and A:T to C:G mutations were predominant in amphibians. If that guess is correct, that would have a high impact, wouldn't it? Why don't the authors pursue that conjecture and get some evidence? As reviewer 1 pointed out, this would require at least introducing the gene into eukaryotic cells and confirming its genomic mutation, which does not seem to be a very difficult experiment. It would also be better to confirm DNA editing activity similar to this enzyme in other amphibians.

Response: Thanks for the suggestion. Our study has demonstrated the mutagenic activity of BjADAT2 in eukaryotic (yeast) cells, and its mutational spectrum using prokaryotic (*E. coli*) cells. Because mutations in genome are generally derived from various mutagenic factors, confirming the relationship between a specific gene and genomic mutations needs overexpression or depletion of the gene in host cell lines. Limited by the lack of amphioxus cell lines at present, it is difficult to evaluate the effects of BjADAT2 on the C:G to T:A and A:T to G:C mutations in amphioxus genome. We will investigate the mutational mechanism in future through high-throughput sequencing and comparing the difference of mutational spectrum between BjADAT2 high-expressed tissues and low-expressed ones in amphioxus. This process will take a relatively long time. Also, it is interesting to test DNA editing activity of ADAT2 in other species of amphioxus, which might shed more light on the mutagenic mechanism of amphioxus ADAT2.

Additional Comments,

From lone 337, the authors described “Several DNA deaminases also edit RNA, such as APOBECs and ADARs.” However, as the name suggests, ADAR is an enzyme that catalyzes RNA and does not work on DNA. APOBECs were also originally discovered as an enzyme that edits apolipoprotein B mRNA in the intestine. The APOBEC family is diverse, with some enzymes catalyzing primarily DNA and others both DNA and RNA, but APOBEC1 is not believed to act on DNA. It should be corrected.

Response: We are sorry for this mistake. This sentence has been corrected as following:

“Several DNA deaminases also edit RNA, such as APOBEC1 (Which targets C-to-U on ssRNA²⁸ and has been applied in both RNA- and DNA-base editors^{29, 30, 31, 32}), and ADAR1 and 2 (Which target A-to-I on dsRNA³³ and have been applied in RNA base editor^{34, 35}).” APOBEC1 can act on both ssDNA and ssRNA (Pecori et al., 2022, *Nat Rev Genet* 23:505-518). The target of ADAR to mismatched adenosines on the DNA strand of DNA/RNA hybrids was reported by Zheng et al., 2017 (*Nucleic acids research* 45:3369-3377). These information have been cited in the paper.

Reviewer #3:

The concerns have been well addressed.

Response: We are grateful to the reviewer for helping us in the improvement of this paper.